# Addressing Disparities in the Propensity Score Distributions for Treatment Comparisons from Observational Studies

**Tingting Zhou** [1,*], **Michael R. Elliott** [2] **and Roderick J. A. Little** [2]

1   U.S. Food and Drug Administration, 10903 New Hampshire Avenue, Silver Spring, MD 20993, USA
2   Department of Biostatistics, School of Public Health, University of Michigan, Ann Arbor, MI 48109, USA
*   Correspondence: tingting.zhou@fda.hhs.gov

**Abstract:** Propensity score (PS) based methods, such as matching, stratification, regression adjustment, simple and augmented inverse probability weighting, are popular for controlling for observed confounders in observational studies of causal effects. More recently, we proposed penalized spline of propensity prediction (PENCOMP), which multiply-imputes outcomes for unassigned treatments using a regression model that includes a penalized spline of the estimated selection probability and other covariates. For PS methods to work reliably, there should be sufficient overlap in the propensity score distributions between treatment groups. Limited overlap can result in fewer subjects being matched or in extreme weights causing numerical instability and bias in causal estimation. The problem of limited overlap suggests (a) defining alternative estimands that restrict inferences to subpopulations where all treatments have the potential to be assigned, and (b) excluding or downweighting sample cases where the propensity to receive one of the compared treatments is close to zero. We compared PENCOMP and other PS methods for estimation of alternative causal estimands when limited overlap occurs. Simulations suggest that, when there are extreme weights, PENCOMP tends to outperform the weighted estimators for ATE and performs similarly to the weighted estimators for alternative estimands. We illustrate PENCOMP in two applications: the effect of antiretroviral treatments on CD4 counts using the Multicenter AIDS cohort study (MACS) and whether right heart catheterization (RHC) is a beneficial treatment in treating critically ill patients.

**Keywords:** causal estimands; PENCOMP; penalized spline; double robustness; multiple imputation; causal inference

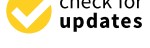



## 1. Introduction

Observational studies for inference about causal effects are valuable when randomization is infeasible or unethical. Valid causal inference in this setting requires adjusting for differences in the distributions of confounders between treatment groups. For example, due to ethical reasons, a randomized controlled trial (RCT) has never been conducted to evaluate the effectiveness of right heart catheterization (RHC), a commonly used procedure in treating critically ill patients. However, because sicker patients are more likely to be treated with RHC and have worse clinical outcomes, appropriate methods are needed to reduce confounding by patient characteristics. To deal with confounding, the propensity score—the probability of treatment assignment as a function of the observed covariates—is often used.

The propensity score summarizes the observed covariates and serves as a dimension-reduction technique. Differences in covariate distributions between treatment groups lead to differences in the average propensity scores between treatment groups [1]. Due to the balancing property of the propensity score, propensity score adjustment can remove the bias due to differences in all observed confounders between treatment groups [2]. Propensity score adjustment methods include matching, stratification, regression adjustment, and inverse probability weighting. The matching and stratification methods construct artificial

controls by matching or grouping treated subjects with non-treated subjects of similar baseline characteristics, i.e., similar propensity scores. Regression-based methods such as penalized spline of propensity prediction (PENCOMP) include a function of propensity score and covariates that are predictive of the outcome. Finally, the inverse probability weighting methods, such as inverse probability of treatment weighting (IPTW) and augmented inverse probability of treatment weighting (AIPTW), control for confounding by weighting subjects by the inverse of the probability of receiving the observed treatments. For these methods to work reliably, there should be sufficient overlap in the propensity score distributions between treatment groups [3]. When there is limited overlap, causal estimates can be imprecise. For example, in matching and stratification, there would be a few or no available subjects in matching or in strata. When the weights are highly variable in presence of limited overlap, IPTW and AIPTW can become numerically unstable. Although PENCOMP is less sensitive to extreme weights [4], being a regression-based method, it can be sensitive to model specification due to extrapolation of regression models.

The most-used estimand for causal comparison is the average treatment effect (ATE), the average treatment effect for the entire population of interest. The ATE estimand is defined as the mean difference in the potential outcomes of two scenarios: when all subjects in the population of interest receive treatment versus when all subjects receive control. However, when there is limited overlap in the propensity score distributions between treatment groups, more credible alternative causal estimands can be estimated by restricting to subpopulations with more covariate balance. Limited overlap can result in extreme propensity scores or extreme weights causing numerical instability and bias in causal estimation. To alleviate the problem of limited overlap, Gutman and Rubin (2013) [5] proposed dropping units outside of the overlap region of estimated propensity scores between treatment groups. Cochran and Rubin (1973) [6] and Dehejia and Wahba (1999) [7] discarded unmatched subjects. Similarly, Rosenbaum (2012) [8] proposed an algorithm for choosing an optimal set of treated subjects, where some treated subjects were dropped due to poor matching quality. Ho et al. (2007) [9] proposed a two-stage approach. In the first stage, all the treated units were paired with their closest control units, and only the matched units were included in the second stage for further adjustment. Crump et al. (2009) [10] and Yoshida et al. (2019) [11] restricted analysis to a subpopulation defined by trimming extreme propensity scores. Li et al. (2017, 2019) [12,13] weighted subjects by the overlap (ATO) weight to balance the weighted distributions of the covariates between treatment groups in a fashion that minimizes the asymptotic variance of the causal effect. Li and Greene (2013) [14] proposed weighting subjects by the match weight. Mao et al. (2018) [15] studied the class of modified inverse probability weighted estimators to address limited overlap.

Restriction of treatment comparison to a subpopulation allows a more precise and credible causal estimate for the subpopulation but changes the causal estimand. Sturmer et al. (2010) [16] argued that extreme propensity scores could be due to some important unmeasured confounding, so restriction of treatment comparisons to subjects within a common range of propensity scores could increase the validity of causal estimate. Imbens and Wooldridge (2009) [17] argued that the focus on the ATE estimand for causal comparison in practice can be unrealistic. Investigators might be less likely to ask patients with extreme propensity scores to participate in a study or receive a treatment (or control) due to excessive risk involved. Thus, the subpopulations obtained by applying more weights to subjects with propensity scores close to 0.5 and less weights to subjects with propensity scores close to 0 or 1 may resemble the population targeted by randomized trials [12,15]. In this paper, we focus our attention on alternative causal estimands and analysis methods to ensure that more robust causal inferences are possible when there is limited overlap in the propensity score distributions between treatment groups. We evaluate the performance of PENCOMP in estimating alternative estimands that restrict causal inference to the subset of the population with sufficient overlap to make robust inference.

The remainder of the paper is structured as follows. In Section 2, we discuss the standard causal estimand (average treatment effect or ATE) along with three alternative estimands designed to deal with overlap. In Section 3, we describe three estimators (PENCOMP along with two weighted estimators) for estimating the estimands described in Section 2. In Section 4, we conduct simulation studies of the performance of the three estimators for the four estimands we consider. In Section 5, we illustrate the methods in two applications: the effect of antiretroviral treatments on CD4 counts and whether right heart catheterization (RHC) is a beneficial treatment in treating critically ill patients. Section 6 concludes with a review of the results and a discussion of possible next steps.

## 2. Causal Estimands

In a study with treatments administered as a single time point, let $X_i$ and $Z_i$ denote the vector of baseline covariates and a binary treatment for subject $i = 1, \cdots, n$, respectively. Let $Z_i = \{0, 1\}$ denote a binary treatment with $Z_i = 1$ for treatment and $Z_i = 0$ for control. Let $Y_i^{Z_i}$ denote the potential outcome under $Z_i$ for subject $i$. Under Rubin's causal model, the treatment effect for a subject $i$ is defined as the difference between the potential outcomes under the two treatments received by the subject. To estimate the causal effects, we make the following three assumptions.

1. Stable Unit Treatment Value Assumption, SUTVA [18]: (a) The observed outcome under the assigned treatment is the same as the potential outcome under that treatment, and (b) the potential outcomes for a given subject are not influenced by the treatment assignments of other subjects [18,19].

2. Positivity: Each subject has a positive probability of being assigned to either treatment of interest. The assumption is violated when there exists neighborhoods in the covariate space where all subjects are assigned the same treatment.

3. Ignorable treatment assignment: $(Y^1, Y^0) \perp\!\!\!\perp Z \mid X$: Treatment assignment is independent of the potential outcomes, given the observed covariates.

We now describe four different target estimands of causal effects: the ATE, the average treatment effect truncated on propensity score, or truncated average treatment effect (TATE), the average treatment effect on an evenly matchable set (ATM), and the average treatment effect on the overlap population (ATO)

### 2.1. ATE

The ATE is the average of the individual level treatment effect defined on the entire population for whom both treatments are appropriate therapies, $E(Y^1 - Y^0)$, which is often used and easy to interpret. This estimand is the population average treatment effect, also denoted as PATE in Imai et al. (2008) [20]. ATE/PATE should be distinguished from the sample average treatment effect (SATE), which is the average over a sample. In this paper, we focus on estimating the treatment effect over a population rather than over a sample. The estimation error of the population level treatment effect from a sample can be decomposed into error due to sample selection and error due to treatment imbalance [20]. By restricting causal estimation to subpopulations to deal with limited overlap, alternative causal estimands are obtained.

### 2.2. TATE

One possible alternative estimand can be defined by truncating extreme propensity scores. For unit $i$ in the population with covariate value $X_i$, let $P_z(X_i) = \Pr(Z_i = z|X_i)$ denote the propensity of receiving treatment $z$, for $z = \{0, 1\}$. The positivity condition holds for the set of units $S(0)$ where $S(0) = \{i : P_z(X_i) > 0\}$. We can restrict inferences to the subpopulation $S(\alpha)$ of $S(0)$, where the probability of all treatment assignments is greater than a pre-defined level of $\alpha$ directly, that is $S(\alpha) = \{i : P_z(X_i) > \alpha, \text{ for } z = \{0, 1\}\}$. We refer to this estimand as the truncated average treatment effect (TATE). The estimand is defined on the true propensity. However, the true propensity is unknown, and for estimation purposes, the truncation is based on an estimated propensity. Thus the quality

of the TATE estimator depends on both the correct specification of the propensity score model and sufficient data to accurately estimate the true propensity scores.

*2.3. ATM*

Samuels (2017) [21] defined the ATM estimand, the average treatment effect on an evenly matchable set $M$, $E(Y^1 - Y^0 \mid M)$. The ATM estimand is targeted by one-to-one pair matching without replacement on propensity score with a caliper, simply called pair matching throughout the paper. A unit is evenly matchable if, within a small propensity score stratum centered around the unit, there are at least as many units from the other group as from its own group. Suppose we divide the range of the propensity score into many small strata. Within each stratum, if there are equal numbers of units from both groups, all the units are evenly matchable; otherwise, only the units from the least prevalent group are evenly matchable. The evenly matched set is the union of all the matchable units from all the strata. Li and Greene (2013) [14] proposed the match weight as an analog to pair matching. Thus, the ATM can also be defined as the weighted average treatment effect $E[\omega(P_1(X_i))\Delta_i]/E[\omega(P_1(X_i))]$, where the weight $\omega(P_1(X_i)) = \min\{P_1(X_i), P_0(X_i)\}$, and $\Delta_i$ is the individual conditional treatment effect for subject $i$ [14].

*2.4. ATO*

Li et al. (2017) [12] defined another estimand, the average treatment effect on the overlap population, ATO. The overlap population is created by down-weighting the units with extreme propensity scores and up-weighting the units with propensity score close to 0.5. The target population is "the units whose combination of characteristics could appear with substantial probability in either treatment group" [12]. The ATO is defined as the weighted average treatment effect $E[\omega(P_1(X_i))\Delta_i]/E[\omega(P_1(X_i))]$, where the weight $\omega(P_1(X_i)) = Z_i P_0(X_i) + (1 - Z_i)P_1(X_i)$, and $\Delta_i$ is the individual conditional treatment effect for subject $i$. Although the targeted population is theoretically more balanced in the covariates between the treated and control groups, it is arguably less interpretable than the original population.

**3. Causal Estimators**

*3.1. IPTW*

Each subject $i$ is weighted by the balancing weight equal to the inverse of the probability of receiving the treatment to which they were assigned: $W_i = \omega(\hat{P}_1(X_i))/\Big\{Z_i\hat{P}_1(X_i) + (1 - Z_i)(1 - \hat{P}_1(X_i))\Big\}$. The weighted estimator for treatment effect $\Delta$ is defined as follows [15]:

$$\hat{\Delta}_{IPTW} = \frac{\sum_{i=1}^{n} W_i Z_i Y_i}{\sum_{i=1}^{n} W_i Z_i} - \frac{\sum_{i=1}^{n} W_i(1 - Z_i)Y_i}{\sum_{i=1}^{n} W_i(1 - Z_i)}$$

Different specifications of $\omega(\hat{P}_1(X_i))$ yield average treatment effects for different subpopulations. For the ATE estimand, $\omega(\hat{P}_1(X_i))$ is 1, which defines the IPTW estimator. For the ATO estimand, $\omega(\hat{P}_1(X_i))$ is $\hat{P}_1(X_i) \times \hat{P}_0(X_i)$. For TATE, $\omega(\hat{P}_1(X_i))$ is set as $I\{i \in S(\alpha)\}$, where $I$ is the indicator. For the ATM estimand, $\omega(\hat{P}_1(X_i))$ is set as $\min\Big(\hat{P}_1(X_i), \hat{P}_0(X_i)\Big)$.

For each estimand, the $\hat{\Delta}_{IPTW}$ is computed on the original data $S$. The standard errors are estimated by bootstrapping the data. The procedure is as follows.

(a) For $d = 1, \cdots, D$, generate a bootstrap sample $S^d$ from the original data $S$ by sampling units with replacement.

(b) For each sample $S^d$, compute the weighted estimator on each bootstrap sample, $\hat{\Delta}_{IPTW}^d$.

(c) Compute the standard error $\widehat{sd}_D$ for $\hat{\Delta}_{IPTW}$ based on $D$ bootstrap samples as follows:

$$\hat{sd}_D^2 = \sum_{d=1}^{D} (\hat{\Delta}_{IPTW}^d - \hat{\Delta}_{\cdot}^*)^2/(D-1)$$

where $\hat{\Delta}_{\cdot}^* = \sum_{d=1}^{D} \hat{\Delta}_{IPTW}^d/D$. The 95% confidence intervals are computed as $\hat{\Delta}_{IPTW} \pm 1.96\hat{sd}_D$.

### 3.2. AIPTW

For each inverse of propensity weighted estimator described in Section 3.1, an augmented inverse of propensity weighted estimator can be defined as follows [15]:

$$\hat{\Delta}_{AIPTW} = \frac{\sum_{i=1}^{n} \omega(\hat{P}_1(X_i))\{m_1(X_i, \alpha_1) - m_0(X_i, \alpha_0)\}}{\sum_{i=1}^{n} \omega(\hat{P}_1(X_i))} + \frac{\sum_{i=1}^{n} W_i Z_i\{Y_i - m_1(X_i, \alpha_1)\}}{\sum_{i=1}^{n} W_i Z_i}$$
$$- \frac{\sum_{i=1}^{n} W_i(1 - Z_i)\{Y_i - m_0(X_i, \alpha_0)\}}{\sum_{i=1}^{n} W_i(1 - Z_i)}$$

where $m_1(X_i, \alpha_1) = E(Y_i|X_i, Z_i = 1)$ and $m_0(X_i, \alpha_0) = E(Y_i|X_i, Z_i = 0)$.

Alternatively, AIPTW can also be constructed as described in Kang & Shafer (2007) [22], as a solution to the following equation:

$$\frac{1}{n} \sum_{i=1}^{n} \hat{U}_i + \frac{1}{n} \sum_{i=1}^{n} Z_i \hat{P}_1(X_i)^{-1}(U_i - \hat{U}_i) = 0,$$

where $U_i = \frac{Y_i - \mu}{\sigma^2}$ and $\hat{U}_i$ is the quasi-score function of $U_i$ with $Y_i$ replaced by $m_1(X_i, \alpha_1)$. In the simulation studies, we use the first version of AIPTW. Similar bootstrap procedures as described in Section 3.1 can be used to estimate the standard error for $\hat{\Delta}_{AIPTW}$.

### 3.3. PENCOMP

PENCOMP is a robust multiple imputation-based approach to causal inference [4,23]. PENCOMP builds on the Penalized Spline of Propensity Prediction method (PSPP) for missing data problems [24,25]. Although PENCOMP was developed to deal with longitudinal settings where later treatment assignments can be confounded by early outcomes, here we consider a simple setting where treatments are assigned only once. Since each subject receives one treatment, we observe the potential outcome under the observed treatment but not the potential outcome under the alternative treatment. Thus, inference about causal effects can be framed as a missing data problem [26,27], where the counterfactual outcome is imputed, and inference is conducted using multiple imputation.

PENCOMP [4,23] applies the idea of PSPP to the causal inference setting, with the propensity of response replaced by the propensity of treatment assignment and the missing data being the outcomes under unassigned treatments. We estimate the propensity to be assigned to each treatment by a regression method suitable for a categorical outcome, for example, by logistic regression if there are two treatments, or polytomous regression if there are more than two treatments. We then predict the potential outcomes for the treatments not assigned to subjects using regression models that include splines on the logit of the propensity to be assigned that treatment and a function of other covariates that are predictive of the outcome. Separate models are fitted for each treatment group. Under SUTVA, positivity and ignorability assumptions stated in Section 2, the marginal mean from the imputation model is consistent if (1) the regression models for the potential outcomes are correctly specified; or, (2) the propensity models are correctly specified, and the relationship between the outcomes and the propensity are correctly specified. The latter assumption can be met under relatively weak conditions by regressing the outcomes on the spline of the logit of the propensity, since the spline does not impose strong assumptions on the functional form of the relationship between the outcomes and the propensity.

PENCOMP in a single time point treatment setting can be implemented as follows:

(a) For $d = 1, \cdots, D$, generate a bootstrap sample $S^d$ from the original data $S$ by sampling units with replacement. Then carry out steps (b)–(d) for each sample $S^d$:

(b) Estimate the propensity score model for the distribution of $Z$ given $X$, with regression parameters $\gamma_z$. The propensity to be assigned treatment $Z = z$ is denoted as $\hat{P}_z(X) = Pr(Z = z | X, \hat{\gamma}_z^d)$, where $\hat{\gamma}_z^d$ is the ML estimate of $\gamma_z$. Define $\widehat{P^*}_z = \log[\hat{P}_z(X)/(1 - \hat{P}_z(X))]$.

(c) For each $z = \{0, 1\}$, using the cases assigned to treatment group $z$, for a continuous outcome, estimate a normal linear regression of $Y^z$ on $X$, with mean

$$E(Y^z | X, Z = z, \theta_z, \beta_z) = s(\widehat{P^*}_z | \theta_z) + g_z(X; \beta_z),$$

For a binary outcome, assume a logistic regression model as follows [28]:

$$logit\{Pr(Y^z = 1 | X, Z = z, \theta_z, \beta_z)\} = s(\widehat{P^*}_z | \theta_z) + g_z(X; \beta_z),$$

where $s(\widehat{P^*}_z | \theta_z)$ denotes a penalized spline with fixed knots [29–31], with parameters $\theta_z$, and $g_z()$ represents a parametric function of covariates predictive of the outcome, including covariates that are adequately balanced by the estimated propensity score models, indexed by parameters $\beta_z$. A different spline function can be fitted for each treatment group.

(d) For $z = 0, 1$, impute $Y^z$ for subjects in treatment group $1 - z$ in the original data set with draws from the predictive distribution of $Y^z$ given $X$ from the regression in (d), with ML estimates $\hat{\theta}_z^{(d)}, \hat{\beta}_z^{(d)}$ substituted for the parameters $\theta_z, \beta_z$, respectively.

(e) Let $\hat{\Delta}^d$ and $V^d$ denote the estimated causal effect and its associated variance in complete dataset $d$. For TATE, we restrict causal comparison to the set of cases defined by $S(\alpha)$ based on the estimated propensity score and calculate the $\hat{\Delta}^d$ and $V^d$ on the restricted sample. For the ATM estimand, we restrict causal comparison to the set of cases remaining after pair matching and calculate $\hat{\Delta}^d$ and $V^d$ based on the matched set. The MI estimate of the treatment effect $\Delta$ is then $\bar{\Delta}_D = \frac{1}{D} \sum_{d=1}^{D} \hat{\Delta}^d$, and the MI estimate of the variance of $\bar{\Delta}_D$ is $T_D = \bar{V}_D + (1 + 1/D) B_D$, where $\bar{V}_D = \sum_{d=1}^{D} V^d / D$, $B_D = \sum_{d=1}^{D} (\hat{\Delta}^d - \bar{\Delta}_D)^2 / (D - 1)$. Then $\Delta$ is $t$ distributed with degree of freedom $v$, $(\Delta - \bar{\Delta}_D) T_D^{\frac{-1}{2}} \sim t_v$, where $v = (D - 1)(1 + \bar{V}_D / ((D + 1) \times B_D))^2$.

As noted by a reviewer, a potential alternative to PENCOMP that could also provide a form of at least approximately doubly robust estimation would be to implement only an outcome model and then use regression matching to estimate counterfactual outcomes [32]. We do not pursue this option here.

## 4. Simulation

### 4.1. Study Design

We conducted simulation studies to assess the finite sample performance of PENCOMP-MI, compared with IPTW and AIPTW in estimating the ATE and alternative estimands when the overlap is low. Our simulation study design considered a single time point with a binary treatment and linear and logistic regression models for continuous and binary outcomes, along with two forms of model misspecification. We considered three sets of models for PENCOMP-MI and AIPTW: (a) correctly specified outcome and propensity score models; (b) a correctly specified outcome model only; and (c) a correctly specified propensity score model only. For IPTW, we considered only a correctly specified or misspecified propensity score model. In each simulation scenario, 500 simulated datasets were created for a sample size of 500. For PENCOMP-MI, 200 complete datasets were created to obtain the estimates, standard errors and 95% confidence intervals (CI). For IPTW and AIPTW, 500 bootstraps were used to estimate the standard errors and 95% CI. For PENCOMP-MI, a truncated linear basis with 20 equally-spaced knots was used. To estimate the ATM estimand using PENCOMP-MI, for each bootstrap sample, we first performed pair matching using propensity score with a caliper of 0.25 times the standard deviation of the logit of estimated propensity score, and then calculated the estimates on the matched set according to the procedures described in Section 3.1.

We compared these methods in terms of empirical bias, root mean squared error (RMSE), and 95% CI (non) coverage. The performance for each method is evaluated according to its target estimand.

Our simulation design is similar to that described in Li et al. (2019) [13]. For each simulated dataset, we first generate six variables $V_1, \cdots, V_6$, from a multivariate normal distribution with zero mean, unit marginal variance and a compound symmetric covariance structure. Let $X_1, \cdots, X_6$ be the observed baseline covariates. The covariates $X_1, \cdots, X_3$ are continuous and equal to $V_1, \cdots, V_3$, respectively. Let $X_4, \cdots, X_6$ be binary covariates defined as $X_4 = I_{V_4 < 0}$, $X_5 = I_{V_5 < 0}$, and $X_6 = I_{V_6 < 0}$. The treatment $Z$ is Bernoulli distributed with a treatment assignment probability as shown below:

$$P(Z = 1) = \{1 + exp[-(\alpha_0 + \alpha_1 X_1 + \alpha_2 X_2 + \alpha_3 X_3 + \alpha_4 X_4 + \alpha_5 X_5 + \alpha_6 X_6)]\}^{-1}$$

where $(\alpha_0, \alpha_1, \alpha_2, \alpha_3, \alpha_4, \alpha_5, \alpha_6) = (0.40, 0.15, 0.3, 0.3, -0.2, -0.25, -0.25)$.

For the continuous outcome, we assume the outcome $Y$ is normally distributed with a variance of 1 and a mean that depends on the observed covariates as shown below:

$$Y \sim N(\beta_0 + \beta_1 X_1 + \beta_2 X_2 + \beta_3 X_3 + \beta_4 X_4 + \beta_5 X_5 + \beta_6 X_6 + \beta_7 Z, 1)$$

where $(\beta_0, \beta_1, \beta_2, \beta_3, \beta_4, \beta_5, \beta_6, \beta_7) = (0, -0.5, -0.5, -1.5, 0.8, 0.8, 1.0, 0.75)$

For the binary outcome, we assume a logistic regression model with coefficients as shown below:

$$Y \sim Bernoulli(\{1 + exp[-(\beta_0 + \beta_1 X_1 + \beta_2 X_2 + \beta_3 X_3 + \beta_4 X_4 + \beta_5 X_5 + \beta_6 X_6 + \beta_7 Z)]\}^{-1})$$

where $(\beta_0, \beta_1, \beta_2, \beta_3, \beta_4, \beta_5, \beta_6, \beta_7) = (0, -1.0, -1.0, -3.0, 1.6, 1.6, 2.0, 2.75)$

We misspecified the propensity score model by omitting the covariates $X_5$ and $X_6$. We misspecified the outcome models for $Y^1$ and $Y^0$ by including only the covariates $X_1$ and $X_2$. In addition, for PENCOMP, we specified another form of outcome misspecification by excluding all covariates, i.e., fitting a penalized spline on the propensity score only. This model specification allowed us to compare PENCOMP with IPTW directly.

The true treatment effects for all the estimands were 0.75 for the continuous outcome scenarios. For the binary outcome scenarios, the treatment effects were heterogeneous. For the heterogeneous treatment effects, we calculated the truth by simulating 200,000 observations and taking the corresponding averages. For the ATE estimand, the truth was taken to be the mean difference in the potential outcomes across the 200,000 observations. For the ATO and ATM estimands, the ATO weights and the match weights were applied to the 200,000 individual treatment effects, respectively, and the truths were taken to be the weighted means. For TATE, the truths were taken to be the mean difference in the potential outcomes in the truncated subpopulation.

### 4.2. Results

Results for the scenario with a homogeneous treatment effect and a continuous outcome are shown in Tables 1–3. Table 1 shows the results for the model specification with correctly specified outcome and propensity score models. The IPTW had a large bias, a large RMSE and a poor coverage for the ATE estimand. Restricting to truncated estimands improved the performance of IPTW. For example, the IPTW targeting the ATE estimand had a bias of 0.555 (74%) and a RMSE of 1.12. The truncated estimand TATE1% reduced the bias to 0.092 (12%) and the RMSE to 0.045. For the ATE and truncated estimands, both PENCOMP and the AIPTW had substantially lower RMSEs than IPTW, with the RMSEs ranging from 0.14 to 0.26. PENCOMP had smaller RMSEs than AIPTW for the ATE and TATE estimands. In addition, PENCOMP with restriction improved the RMSEs slightly. For example, PENCOMP had a RMSE of 0.20 for the ATE estimand and 0.14 for the TATE5% estimand. PENCOMP had a slight overcoverage, with 95% CI non-coverage of 2.2 for the ATE estimand.

**Table 1.** Empirical bias (in absolute and percentage), RMSE, and 95% CI (non) coverage, across the methods in a linear and continuous outcome model for the scenario where both the propensity score model and the outcome model were correctly specified. Results were based on 500 simulations with sample size of 500; 500 bootstraps were used to estimate standard errors for the weighted estimators and 200 complete datasets for PENCOMP. ATE = Average treatment effect; ATM = average treatment effect on an evenly matchable set; ATO = average treatment effect on the overlap population; TATE$\alpha$ = truncated average treatment effect with a truncation at a pre-defined $\alpha$ level.

| Estimand | Estimator | Truth $\times$1000 | Absolute Bias $\times$1000 | Percent Bias $\times$100 | RMSE $\times$100 | Non Coverage $\times$100 |
|---|---|---|---|---|---|---|
| ATE | IPTW | 750 | 555 | 74 | 112 | 41.4 |
| ATE | AIPTW | 750 | 3 | 0 | 26 | 9.2 |
| ATE | PENCOMP | 750 | 6 | 1 | 20 | 2.2 |
| ATM | IPTW | 750 | 0 | 0 | 14 | 5.0 |
| ATM | AIPTW | 750 | 2 | 0 | 14 | 4.2 |
| ATM | PENCOMP | 750 | 3 | 0 | 14 | 2.4 |
| ATO | IPTW | 750 | 4 | 1 | 14 | 5.4 |
| ATO | AIPTW | 750 | 4 | 1 | 14 | 5.2 |
| ATO | PENCOMP | 750 | 3 | 0 | 14 | 4.0 |
| TATE0.01 | IPTW | 750 | 92 | 12 | 45 | 10.0 |
| TATE0.01 | AIPTW | 750 | 13 | 2 | 21 | 8.8 |
| TATE0.01 | PENCOMP | 750 | 5 | 1 | 16 | 4.0 |
| TATE0.05 | IPTW | 750 | 22 | 3 | 22 | 4.8 |
| TATE0.05 | AIPTW | 750 | 12 | 2 | 17 | 4.2 |
| TATE0.05 | PENCOMP | 750 | 5 | 1 | 14 | 4.4 |

Table 2 shows the results for the model specification with a misspecified outcome model only. For the ATE estimand, AIPTW had a bias of 0.229 (31%) with a RMSE of 0.63. PENCOMP had a much smaller bias of 0.044 (6%) and a RMSE of 0.22. PENCOMP also had a 95% non-coverage rate of 2.0%, while AIPTW had a non-coverage of 30.8%. In this case, PENCOMP* with a null outcome model seemed to perform similarly to PENCOMP with a misspecified outcome model, in terms of bias, RMSEs and coverage. In general, PENCOMP seemed to have conservative (over) coverage for the ATE estimand while AIPTW had under coverage. Restricting PENCOMP to the matched set or truncated subpopulations achieved similar RMSEs as those achieved by using ATO and ATM weights. For example, for TATE5%, PENCOMP had a RMSE 0.14 and a 95% non-coverage rate of 3.4%, compared to a RMSE of 0.14 and a 95% non-coverage rate of 3.4% for using the ATO weights. However, compared to using the ATO and ATM weights, PENCOMP had larger biases, when including an incorrectly specified outcome model.

Table 3 shows the results for the model specification with a misspecified propensity score model. When the propensity score model was misspecified, IPTW for all the estimands had large empirical biases, RMSEs and low coverage rates. Both PENCOMP and AIPTW performed much better than IPTW for all the estimands because the outcome models were correctly specified. Misspecifying the propensity score models seemed to alleviate the problem of extreme propensity score and slightly improved the performance of AIPTW in this case. The performance of PENCOMP did not change much as PENCOMP was less affected by extreme weights, compared to IPTW and AIPTW.

**Table 2.** Empirical bias (in absolute and percentage), RMSE, and 95% CI (non) coverage, across the methods in a linear and continuous outcome model for the scenario where the outcome models were misspecified. Results were based on 500 simulations with sample size of 500; 500 bootstraps were used to estimate standard errors for the weighted estimators and 200 complete datasets for PENCOMP. PENCOMP* denotes the outcome misspecification with a null outcome model by fitting a penalized spline on the propensity score only. ATE = Average treatment effect; ATM = average treatment effect on an evenly matchable set; ATO = average treatment effect on the overlap population; TATE$\alpha$ = truncated average treatment effect with a truncation at a pre-defined $\alpha$ level.

| Estimand | Estimator | Truth ×1000 | Absolute Bias ×1000 | Percent Bias ×100 | RMSE ×100 | Non Coverage ×100 |
|---|---|---|---|---|---|---|
| ATE | IPTW | 750 | 555 | 74 | 112 | 41.4 |
| ATE | AIPTW | 750 | 229 | 31 | 63 | 30.8 |
| ATE | PENCOMP | 750 | 44 | 6 | 22 | 2.0 |
| ATE | PENCOMP* | 750 | 23 | 3 | 22 | 1.2 |
| ATM | IPTW | 750 | 0 | 0 | 14 | 5.0 |
| ATM | AIPTW | 750 | 3 | 0 | 15 | 4.6 |
| ATM | PENCOMP | 750 | 29 | 4 | 14 | 1.8 |
| ATM | PENCOMP* | 750 | 47 | 6 | 15 | 1.2 |
| ATO | IPTW | 750 | 4 | 1 | 14 | 5.4 |
| ATO | AIPTW | 750 | 6 | 1 | 14 | 5.4 |
| ATO | PENCOMP | 750 | 19 | 3 | 14 | 3.4 |
| ATO | PENCOMP* | 750 | 37 | 5 | 14 | 2.6 |
| TATE0.01 | IPTW | 750 | 92 | 12 | 45 | 10.0 |
| TATE0.01 | AIPTW | 750 | 49 | 7 | 35 | 11.0 |
| TATE0.01 | PENCOMP | 750 | 20 | 3 | 16 | 2.8 |
| TATE0.01 | PENCOMP* | 750 | 30 | 4 | 16 | 2.2 |
| TATE0.05 | IPTW | 750 | 22 | 3 | 22 | 4.8 |
| TATE0.05 | AIPTW | 750 | 16 | 2 | 21 | 3.6 |
| TATE0.05 | PENCOMP | 750 | 17 | 2 | 14 | 3.4 |
| TATE0.05 | PENCOMP* | 750 | 36 | 5 | 15 | 2.2 |

Tables 4–6 show the results for the binary outcome scenarios. Table 4 shows the results for the model specification with correctly specified outcome and propensity score models. Table 5 shows the results for the scenario with a misspecified outcome model only and Table 6 for the scenario with a misspecified propensity score model. The same patterns were observed in all three scenarios. AIPTW and PENCOMP targeting the ATE estimand had the least RMSEs. The alternative estimands had higher biases and RMSEs, regardless of the methods. In general, PENCOMP tended to have comparable or smaller RMSEs than AIPTW for both the ATE and restricted estimands. For example, when the outcome model was misspecified, PENCOMP targeting the ATE estimand had a RMSE of 0.024, while AIPTW had a RMSE of 0.046. Furthermore, fitting a penalized spline on the propensity score model (without a parametric outcome model) had better performance than weighting by the inverse of the propensity score in terms of bias, RMSEs and coverage.

**Table 3.** Empirical bias (in absolute and percentage), RMSE, and 95% CI (non) coverage, across the methods in a linear and continuous outcome model for the scenario where the propensity score models were misspecified. Results were based on 500 simulations with sample size of 500; 500 bootstraps were used to estimate standard errors for the weighted estimators and 200 complete datasets for PENCOMP. ATE = Average treatment effect; ATM = average treatment effect on an evenly matchable set; ATO = average treatment effect on the overlap population; TATE$\alpha$ = truncated average treatment effect with a truncation at a pre-defined $\alpha$ level.

| Estimand | Estimator | Truth ×1000 | Absolute Bias ×1000 | Percent Bias ×100 | RMSE ×100 | Non Coverage ×100 |
|---|---|---|---|---|---|---|
| ATE | IPTW | 750 | 963 | 128 | 127 | 62.2 |
| ATE | AIPTW | 750 | 9 | 1 | 24 | 8.0 |
| ATE | PENCOMP | 750 | 0 | 0 | 20 | 2.8 |
| ATM | IPTW | 750 | 435 | 58 | 46 | 74.8 |
| ATM | AIPTW | 750 | 2 | 0 | 14 | 5.0 |
| ATM | PENCOMP | 750 | 2 | 0 | 14 | 2.6 |
| ATO | IPTW | 750 | 439 | 59 | 47 | 78.0 |
| ATO | AIPTW | 750 | 3 | 0 | 14 | 6.0 |
| ATO | PENCOMP | 750 | 2 | 0 | 14 | 4.4 |
| TATE0.01 | IPTW | 750 | 552 | 74 | 69 | 39.0 |
| TATE0.01 | AIPTW | 750 | 1 | 0 | 20 | 8.2 |
| TATE0.01 | PENCOMP | 750 | 1 | 0 | 16 | 5.2 |
| TATE0.05 | IPTW | 750 | 461 | 61 | 51 | 54.8 |
| TATE0.05 | AIPTW | 750 | 0 | 0 | 16 | 5.2 |
| TATE0.05 | PENCOMP | 750 | 2 | 0 | 14 | 4.4 |

**Table 4.** Empirical bias (in absolute and percentage), RMSE, and 95% CI (non) coverage, across the methods for a binary outcome in the scenario where both the propensity score and the outcome models were correctly specified. Results were based on 500 simulations with sample size of 500; 500 bootstraps were used to estimate standard errors for the weighted estimators and 200 complete datasets for PENCOMP. ATE = Average treatment effect; ATM = average treatment effect on an evenly matchable set; ATO = average treatment effect on the overlap population; TATE$\alpha$ = truncated average treatment effect with a truncation at a pre-defined $\alpha$ level.

| Estimand | Estimator | Truth ×1000 | Absolute Bias ×1000 | Percent Bias ×100 | RMSE ×100 | Non Coverage ×100 |
|---|---|---|---|---|---|---|
| ATE | IPTW | 144 | 33.07 | 22.90 | 8.8 | 19.8 |
| ATE | AIPTW | 144 | 2.02 | 1.40 | 2.7 | 3.8 |
| ATE | PENCOMP | 144 | 0.44 | 0.31 | 2.4 | 3.2 |
| ATM | IPTW | 252 | 4.67 | 1.85 | 5.1 | 5.4 |
| ATM | AIPTW | 252 | 3.95 | 1.57 | 4.7 | 3.8 |
| ATM | PENCOMP | 252 | 6.18 | 2.45 | 4.7 | 2.4 |
| ATO | IPTW | 242 | 4.57 | 1.89 | 4.8 | 4.2 |
| ATO | AIPTW | 242 | 4.17 | 1.72 | 4.3 | 3.4 |
| ATO | PENCOMP | 242 | 4.11 | 1.70 | 4.4 | 0.0 |
| TATE0.01 | IPTW | 177 | 2.94 | 1.66 | 6.0 | 5.6 |
| TATE0.01 | AIPTW | 177 | 6.46 | 3.65 | 3.5 | 2.8 |
| TATE0.01 | PENCOMP | 177 | 8.60 | 4.86 | 3.4 | 3.4 |
| TATE0.05 | IPTW | 231 | 5.59 | 2.42 | 5.3 | 4.0 |
| TATE0.05 | AIPTW | 231 | 6.42 | 2.78 | 4.4 | 3.6 |
| TATE0.05 | PENCOMP | 231 | 6.76 | 2.92 | 4.2 | 3.2 |

**Table 5.** Empirical bias (in absolute and percentage), RMSE, and 95% CI (non) coverage, across the methods for a binary outcome in the scenario where the outcome models were misspecified. Results were based on 500 simulations with sample size of 500; 500 bootstraps were used to estimate standard errors for the weighted estimators and 200 complete datasets for PENCOMP. PENCOMP* denotes the outcome misspecification with a null outcome model by fitting a penalized spline on the propensity score only. ATE = Average treatment effect; ATM = average treatment effect on an evenly matchable set; ATO = average treatment effect on the overlap population; TATE$\alpha$ = truncated average treatment effect with a truncation at a pre-defined $\alpha$ level.

| Estimand | Estimator | Truth $\times 1000$ | Absolute Bias $\times 1000$ | Percent Bias $\times 100$ | RMSE $\times 100$ | Non Coverage $\times 100$ |
|---|---|---|---|---|---|---|
| ATE | IPTW | 144 | 33.07 | 22.90 | 8.8 | 19.8 |
| ATE | AIPTW | 144 | 1.89 | 1.31 | 4.6 | 4.6 |
| ATE | PENCOMP | 144 | 1.02 | 0.71 | 2.4 | 2.2 |
| ATE | PENCOMP* | 144 | 1.22 | 0.84 | 2.5 | 1.8 |
| ATM | IPTW | 252 | 4.67 | 1.85 | 5.1 | 5.4 |
| ATM | AIPTW | 252 | 3.65 | 1.45 | 5.0 | 4.4 |
| ATM | PENCOMP | 252 | 2.25 | 0.89 | 4.7 | 2.0 |
| ATM | PENCOMP* | 252 | 0.67 | 0.27 | 4.7 | 1.2 |
| ATO | IPTW | 242 | 4.57 | 1.89 | 4.8 | 4.2 |
| ATO | AIPTW | 242 | 3.69 | 1.52 | 4.8 | 4.4 |
| ATO | PENCOMP | 242 | 2.22 | 0.92 | 4.3 | 0.0 |
| ATO | PENCOMP* | 242 | 1.45 | 0.60 | 4.3 | 0.0 |
| TATE0.01 | IPTW | 177 | 2.94 | 1.66 | 6.0 | 5.6 |
| TATE0.01 | AIPTW | 177 | 3.83 | 2.17 | 5.1 | 3.4 |
| TATE0.01 | PENCOMP | 177 | 6.86 | 3.88 | 3.3 | 2.0 |
| TATE0.01 | PENCOMP* | 177 | 6.76 | 3.82 | 3.3 | 1.6 |
| TATE0.05 | IPTW | 231 | 5.59 | 2.42 | 5.3 | 4.0 |
| TATE0.05 | AIPTW | 231 | 5.83 | 2.52 | 5.2 | 2.4 |
| TATE0.05 | PENCOMP | 231 | 4.93 | 2.13 | 4.1 | 2.2 |
| TATE0.05 | PENCOMP* | 231 | 4.85 | 2.10 | 4.1 | 2.0 |

**Table 6.** Empirical bias (in absolute and percentage), RMSE, and 95% CI (non) coverage, across the methods for a binary outcome in the scenario where the propensity score models were misspecified. Results were based on 500 simulations with sample size of 500; 500 bootstraps were used to estimate standard errors for the weighted estimators and 200 complete datasets for PENCOMP. ATE = Average treatment effect; ATM = average treatment effect on an evenly matchable set; ATO = average treatment effect on the overlap population; TATE$\alpha$ = truncated average treatment effect with a truncation at a pre-defined $\alpha$ level.

| Estimand | Estimator | Truth $\times 1000$ | Absolute Bias $\times 1000$ | Percent Bias $\times 100$ | RMSE $\times 100$ | Non Coverage $\times 100$ |
|---|---|---|---|---|---|---|
| ATE | IPTW | 144 | 73.89 | 51.17 | 10.8 | 41.8 |
| ATE | AIPTW | 144 | 1.03 | 0.72 | 2.8 | 5.4 |
| ATE | PENCOMP | 144 | 0.11 | 0.078 | 2.5 | 2.4 |
| ATM | IPTW | 242 | 62.99 | 26.01 | 7.9 | 24.4 |
| ATM | AIPTW | 242 | 4.57 | 1.89 | 4.4 | 4.0 |
| ATM | PENCOMP | 242 | 10.72 | 4.43 | 4.6 | 2.0 |
| ATO | IPTW | 231 | 59.58 | 25.81 | 7.6 | 23.8 |
| ATO | AIPTW | 231 | 4.59 | 1.99 | 4.0 | 3.4 |
| ATO | PENCOMP | 231 | 4.94 | 2.14 | 4.1 | 0.0 |
| TATE0.01 | IPTW | 169 | 54.94 | 32.55 | 7.9 | 22.4 |
| TATE0.01 | AIPTW | 169 | 4.62 | 2.74 | 3.2 | 4.2 |
| TATE0.01 | PENCOMP | 169 | 5.96 | 3.53 | 3.0 | 3.2 |
| TATE0.05 | IPTW | 214 | 56.76 | 26.49 | 7.6 | 18.6 |
| TATE0.05 | AIPTW | 214 | 5.80 | 2.71 | 4.0 | 3.2 |
| TATE0.05 | PENCOMP | 214 | 6.33 | 2.96 | 3.8 | 2.6 |

## 5. Application

### 5.1. Multicenter AIDS Cohort Study (MACS)

The Multicenter AIDS Cohort study (MACS) was started in 1984 [33]. A total of 4954 men were enrolled into the study and followed semi-annually. At each visit, data from physical examination, medical and behavioral history, and blood test results were collected. The primary outcome of interest was the CD4 count, a continuous measure of how well the immune system functions. We used this dataset to analyze the short-term effects (one year) of using antiretroviral treatments for HIV+ subjects. Here we restricted our analyses to visit 13 when a new questionnaire on antiretroviral treatments was introduced. Treatment was coded to 1 if the patient reported taking any antiretroviral treatments during the past one year according to the new questionnaire. For the analyses, we used a complete case analysis. We transformed the blood counts by taking the square root because the covariate distributions were very skewed.

Let $t = 1, 2$, and 3 represent visit 12, 13 and 14, respectively. Let $X(t = 1, 2)$ denote the blood counts at visit 12 and 13, respectively. Let $Z$ be the binary treatment indicator that takes the value of 1 if the subject took treatments during the periods between visit 12 and visit 13, or between visit 13 and 14 according to the questionnaire. Let $Y(t = 3)$ be the CD4 count one year later. For the propensity score model, we considered blood counts-CD4, CD8, white blood cell (WBC), red blood cell (RBC), and platelets from the most recent 2 visits, as well as demographic variables-age and race. The treatment assignment $Z$ was modeled using a logistic regression. The outcome model was modeled using a normal linear model with CD4, CD8, white blood cell (WBC), red blood cell (RBC), and platelets from the most recent 2 visits and age. A total of 20 equally spaced knots and a truncated linear spline were used.

As shown in Figure 1, the propensity score distributions were very skewed. The treated had propensity of treatment close 1 and the control close to 0. There were 32% of the controls who had estimated propensity scores between the 5% and 95% quantiles of the propensity score distribution of the treated. To check the balance of each covariate between the treated and control groups, we regressed the covariate on the spline of the logit of estimated propensity scores and compared the residuals between the two groups using the *t* statistics and the standardized mean difference. Table A1 in the supplement shows that after adjusting for the propensity scores, the standardized mean differences and the *t* statistics were reduced dramatically.

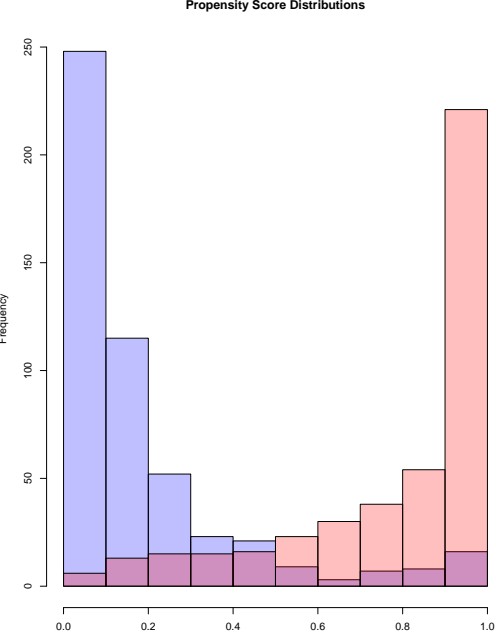

**Figure 1.** Propensity score distributions for the treated and the controls.

We estimated the short-term effect of one year of antiretroviral treatment on CD4 count using PENCOMP, IPTW and AIPTW. The results are summarized in Table 7. The standard errors were obtained using 2000 bootstrap samples. For PENCOMP, 2000 complete datasets were created. The naive estimators were negative, suggesting a harmful effect of antiretroviral treatment on CD4 count, because sicker subjects with lower CD4 counts were more likely to be assigned to treatment. All the causal effect estimates suggested favorable or less harmful effects. When the weights were variable, the PENCOMP estimate for the ATE estimand had a smaller standard error and a narrower 95% confidence interval length than the IPTW and AIPTW estimates. For the ATM and ATO estimands, PENCOMP had comparable standard errors to the AIPTW but had wider confidence interval. Restricting PENCOMP to a truncated subpopulation decreased the SE and 95% confidence interval length.

**Table 7.** MACS dataset: estimated one year change in CD4 count obtained using different methods. Standard errors were calculated based on 2000 bootstrap samples. ATE = Average treatment effect; ATM = average treatment effect on an evenly matchable set; ATO = average treatment effect on the overlap population; TATE$\alpha$ = truncated average treatment effect with a truncation at a pre-defined $\alpha$ level.

| Estimand | Estimator | Estimate | SE | 95% CI Length |
|----------|-----------|----------|-----|---------------|
| ATE | IPTW | 3.66 | 1.78 | 6.97 |
| ATE | AIPTW | −0.18 | 0.70 | 2.75 |
| ATE | PENCOMP | −0.16 | 0.44 | 1.72 |
| ATM | IPTW | −0.10 | 0.34 | 1.34 |
| ATM | AIPTW | 0.24 | 0.32 | 1.24 |
| ATM | PENCOMP | 0.14 | 0.39 | 1.54 |
| ATO | IPTW | 0.16 | 0.30 | 1.16 |
| ATO | AIPTW | 0.20 | 0.31 | 1.20 |
| ATO | PENCOMP | 0.03 | 0.32 | 1.25 |
| TATE0.05 | IPTW | 1.67 | 1.20 | 4.68 |
| TATE0.05 | AIPTW | 0.10 | 0.52 | 2.03 |
| TATE0.05 | PENCOMP | −0.02 | 0.34 | 1.34 |

*5.2. Right Heart Catheterization (RHC)*

Right heart catheterization (RHC) is a common but invasive procedure when treating critically ill patients. Many cardiologists and critical care physicians believe that RHC is beneficial. Due to ethical reasons, randomized controlled trials (RCTs) were never conducted to confirm its effectiveness. Without RCT, observational studies were often used to evaluate its effectiveness. Because sicker patients were more likely to be treated with RCH and to have adverse outcomes, treatment selection in observational studies was often confounded by patient characteristics that influenced the outcomes. Connors et al. (2001) [34] used data from the Study to Understand Prognoses and Preferences for Outcomes and Risks of Treatments (SUPPORT) to assess the effectiveness of RHC. Connors et al. (2001) [34] used propensity score matching to reduce confounding by matching patients without RHC to similar patients with RHC who had the same disease category and concluded that patients with RHC had an increased 30-day mortality, contrary to the common belief of clinical effectiveness.

The SUPPORT study has been reanalyzed by other researchers [10,12,35,36]. Here we applied our method to reanalyze the data and compared to what other researchers have concluded. The study included a total of 5735 critically ill adult patients from five US teaching hospitals between 1989 and 1994. A total of 2184 patients received RHC within 24 h of admission and 3551 patients did not. The treatment was a binary variable, taking the value of 1 if the subject had RHC within the first 24 h after hospitalization. The outcome was a binary variable indicating survival at Day 30.

We used the same propensity score model as described in Li et al. (2018) [12], Hirano and Imbens (2001) [35], Crump et al. (2009) [10] and Traskin and Small (2011) [36]. The model included age, sex, race (black, white, other), years of education, income, type of medical insurance (private, Medicare, Medicaid, private and Medicare, Medicare and Medicaid, or none), primary disease category, secondary disease category, 12 categories of admission diagnosis, ADL and DASI 2 weeks before admission, do-not-resuscitate status on day 1, cancer (none, localized, metastatic), SUPPORT model estimate of the probability of surviving 2 months, acute physiology component of the APACHE III score, Glasgow Coma Score, weight, temperature, mean blood pressure, respiratory rate, heart rate, PaO2/FIO2 ratio, PaCO2, pH, WBC count, hematocrit, sodium, potassium, creatinine, bilirubin, albumin, urine output, and 13 categories of comorbid illness. We included the same set of covariates in the outcome model as those in the propensity score model. Again, a total of 20 equally spaced knots and a truncated linear spline were used.

Table 8 shows the results for different estimands. All the estimates suggest that RHC use led to a higher mortality rate. When targeting the ATE estimand, PENCOMP had a similar estimate to IPTW and AIPTW and had a smaller standard error (SE) and a narrower 95% confidence interval length than IPTW and AIPTW. Restricting PENCOMP to truncated populations achieved similar results to those achieved by using ATO and ATM weights. However, targeting the ATM and ATO estimands, PENCOMP had larger SEs and confidence intervals compared to the IPTW and AIPTW.

**Table 8.** RHC dataset: Estimated log odds ratio of death obtained using different methods. Standard errors were calculated based on 1000 bootstrap samples. ATE = Average treatment effect; ATM = average treatment effect on an evenly matchable set; ATO = average treatment effect on the overlap population; TATE$\alpha$ = truncated average treatment effect with a truncation at a pre-defined $\alpha$ level.

| Estimand | Estimator | Estimate $\times 10^2$ | SE $\times 10^2$ | 95% CI Length $\times 10^2$ |
|---|---|---|---|---|
| ATE | IPTW | 5.84 | 1.69 | 6.63 |
| ATE | AIPTW | 6.51 | 1.58 | 6.21 |
| ATE | PENCOMP | 6.55 | 1.46 | 5.73 |
| ATM | IPTW | 6.52 | 1.39 | 5.44 |
| ATM | AIPTW | 6.80 | 1.39 | 5.45 |
| ATM | PENCOMP | 6.44 | 1.50 | 5.87 |
| ATO | IPTW | 6.53 | 1.36 | 5.32 |
| ATO | AIPTW | 6.72 | 1.36 | 5.34 |
| ATO | PENCOMP | 6.47 | 2.16 | 8.45 |
| TATE0.05 | IPTW | 6.26 | 1.54 | 6.05 |
| TATE0.05 | AIPTW | 6.31 | 1.49 | 5.84 |
| TATE0.05 | PENCOMP | 6.38 | 1.37 | 5.36 |

## 6. Discussion

The restricted estimands considered in this paper are defined based on propensity scores. The subpopulations targeted by the restricted estimands are arguably harder to interpret and not clearly defined in terms of the observed covariates. When there is limited overlap in the propensity score distributions, reporting more precise causal estimates for some subpopulations can be more informative than reporting an imprecise causal estimate for the entire population. However, there is a trade-off between internal and external validity since restriction of causal comparison to some subpopulations also limits the generalizability of the results. Applied researchers have to decide which estimand is more appropriate given the research questions at hand. For example, in the presence of treatment heterogeneity, looking at the restricted estimands could also allow researchers to assess whether there are subpopulations that could benefit from some treatment.

Discarding subjects can reduce the effective sample size and increase the variance of causal estimate. When there is limited overlap, restriction of causal comparison to

some subpopulations with more covariate balance decreases the variance of causal effects. However, the restricted estimands deviate from the ATE estimand when treatment effects are heterogeneous. As the sample size increases, there are often more observed subjects in the treatment and control groups with similar propensity scores and causal effects can be estimated more accurately. Therefore, the range of propensities where the causal effects can be estimated should depend on sample size. If we allow the truncation threshold $\alpha$ to decrease as sample size increases, the truncated estimands approach the ATE estimand. On the contrary, the ATO and the ATM estimands are fixed and might be less relevant when the sample size is large and the target estimand is the ATE.

Simulation results show that the performance of all estimation methods can be improved by defining restricted estimands when the overlap in propensity score distributions is low. When there are extreme weights, PENCOMP tends to outperform the weighted estimators for ATE and perform similarly for restricted estimands. The weighted estimators tend to undercover when there are extreme weights. PENCOMP tends to overcover more, as fitting separate splines by treatment groups could increase variance. For example, when the relationship between outcome and propensity score is linear, fitting separate spline models could be too conservative. Nevertheless, when the weights are extreme, fitting a penalized spline on the propensity score as seen in PENCOMP with a null outcome model can perform much better than just weighting subjects by the inverse of the propensity score. PENCOMP is a viable alternative to the weighted estimators, especially when the ATE is the estimand of interest in the presence of variable weights. To improve the performance of PENCOMP, truncated estimands seem to be better alternative estimands than ATO or ATM estimand. When ATO and ATM are the alternative estimands of interest, the weighted estimators seem to perform better than PENCOMP, as including a misspecified outcome model does not necessarily improve the estimation and including a spline could increase variance in small samples.

Several extensions of our work are possible. Here we have focused on complete case analysis. However, in many practical settings missing data and/or truncation by death is present. In such situations, multiple imputation is often used. PENCOMP, being a multiple imputation procedure, can be built into the multiple imputation procedure for missing data.

Another issue for further exploration is variable selection in the development of the propensity scores. Early developers of the propensity score have argued that all potential pre-treatment potential confounders should be included in the propensity model to avoid "data snooping" and better approximate a randomized trial, where randomization occurs prior to observing the outcomes [37]. On the other hand, including strong predictors of the treatment that are not predictors of the outcome can inflate the variance of the causal estimate [38]. Furthermore, the propensity score plays an important role in identifying the overlap region. Including predictors of the treatment that are not predictors of the outcome could unnecessarily shrink the overlap region in the propensity score distributions. Low overlap then results in fewer matched subjects when matching methods are used or extreme weights when weighting methods are used or shrinks the subset of the population about which can make inference using the approaches discussed in this paper. While previous work has shown that PENCOMP is more stable than competing doubly-robust weighted estimators when non-confounding variables are included in the propensity score model [23], none of these approaches will deal with shrinking overlap. Exploring the tradeoff between risking bias due to inappropriate exclusion of true confounders and inefficiency due to the inclusion of strong predictors of the treatment only remains an open area for research in future studies.

## 7. Disclaimer

This paper represents the opinions of the authors solely. It does not represent the position of the Food and Drug Administration nor the opinions of its members.

**Author Contributions:** Conceptualization, T.Z., M.R.E. and R.J.A.L.; methodology, T.Z., M.R.E. and R.J.A.L.; software, T.Z.; validation, T.Z.; formal analysis, T.Z.; investigation, T.Z., M.R.E. and R.J.A.L.;

resources, T.Z., M.R.E. and R.J.A.L.; data curation, T.Z.; writing—original draft preparation, T.Z.; writing—review and editing, T.Z., M.R.E. and R.J.A.L.; visualization, T.Z.; supervision, M.R.E. and R.J.A.L.; project administration, T.Z., M.R.E. and R.J.A.L.; funding acquisition, Not applicable. All authors have read and agreed to the published version of the manuscript.

**Funding:** This research received no external funding.

**Institutional Review Board Statement:** Not applicable.

**Informed Consent Statement:** Not applicable.

**Data Availability Statement:** Our R code and the used application data sets can be requested by email to the corresponding author.

**Acknowledgments:** The authors thank the Multicenter AIDS Cohort Study (MACS) and SUPPORT study for providing us the datasets for analyses.

**Conflicts of Interest:** The authors declare no potential conflict of interests.

## Appendix A. Supplementary Results from Applications

**Table A1.** MACS dataset: Balance of covariates between the treated and the control groups. We regressed each covariate on the spline of the logit of the propensity of treatment; a truncated linear basis with 10 equally spaced knots was used.

| Covariates | Before Adjusting | | | | After Adjusting | |
|---|---|---|---|---|---|---|
| | Mean Treated | Mean Control | Standardized Mean Difference | T Stats | Standardized Mean Difference | T Stats |
| CD4 visit 12 | 17.15 | 23.97 | −1.13 | 17.32 | 0.0086 | −0.13 |
| CD4 visit 13 | 17.01 | 23.65 | −0.99 | 15.09 | −0.0072 | 0.11 |
| CD8 visit 12 | 30.46 | 31.16 | −0.094 | 1.43 | −0.015 | 0.23 |
| CD8 visit 13 | 29.53 | 30.34 | −0.11 | 1.61 | −0.012 | 0.18 |
| WBC visit 12 | 67.03 | 74.33 | −0.68 | 10.38 | 0.00013 | −0.0020 |
| WBC visit 13 | 65.61 | 72.18 | −0.59 | 8.94 | −0.030 | 0.46 |
| RBC visit 12 | 1.99 | 2.18 | −1.30 | 19.45 | 0.012 | −0.18 |
| RBC visit 13 | 1.93 | 2.18 | −1.96 | 29.65 | −0.018 | 0.28 |
| Platelet visit 12 | 14.76 | 15.03 | −0.12 | 1.75 | −0.0044 | 0.067 |
| Platelet visit 13 | 14.57 | 14.69 | −0.054 | 0.82 | −0.019 | 0.28 |
| age | 39.78 | 38.11 | 0.24 | −3.65 | 0.00062 | −0.0095 |
| white | 0.94 | 0.85 | 0.28 | −4.33 | 0.011 | −0.17 |

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
