# Peer review of "Addressing Disparities in the Propensity Score Distributions for Treatment Comparisons from Observational Studies"

_stats, doi:10.3390/stats5040076_

Round 1

Reviewer 1 Report

The submitted manuscript (ms) studies different propensity score (PS) methods for different target estimands that are defined in situations of limited overlap between treated and control units. In general, I found the ms well-structured. Although the paper only reviews already existing methods in the literature, it is certainly useful to applied researchers.
Major comments:
1.    The Imai et al. (2008) reference should also be discussed regarding different definitions of the treatment effect.
2.    Section 3.2: The augmented inverse of propensity weighted estimator (AIPWT) could be embedded into the literature on doubly robust estimators (e.g., Kang & Schafer, 2007).
3.    Simulation Study: My main concern is that approaches only based on the outcome model are neglected. In the AIPWT estimator, we have two ingredients: the PS model and the outcome model. Hence, it seems plausible that one could opt only for the outcome model and use regression matching (also referred to as regression estimation; see, e.g., Schafer & Kang, 2008). This method should be included in the simulation studies.
Minor comments:
4.    P. 1, Abstract type “study(MACS)”

References:
Imai, K., King, G. and Stuart, E.A., 2008. Misunderstandings between experimentalists and observationalists about causal inference. Journal of the royal statistical society: series A (statistics in society), 171(2), pp.481-502.

Kang, J.D. and Schafer, J.L., 2007. Demystifying double robustness: A comparison of alternative strategies for estimating a population mean from incomplete data. Statistical science, 22(4), pp.523-539.

Schafer, J.L. and Kang, J., 2008. Average causal effects from nonrandomized studies: a practical guide and simulated example. Psychological methods, 13(4), p.279.

Reviewer 2 Report

Please, find my comments in the attached report.

Round 2

Reviewer 1 Report

The revised manuscript investigates different propensity score (PS) methods for different target estimands defined in situations of limited overlap between treated and control units.
Unfortunately, the authors decided not to address my previous Comment 3. I asked to include treatment effect estimators based only on the outcome model. I think that time a “short deadline” is by no means an excuse for not realizing my request. I have two alternative suggestions: (1) include the outcome-based treatment effect estimates as requested, or (2) remove the doubly robust estimators. In the latter case, it is clear that the authors would only believe in the correctness of the treatment assignment model. However, I generally dislike papers that include doubly robust estimators that also allow the possibility of misspecified treatment models but do not include purely outcome model-based approaches.